# The effects of postoperative treadmill exercise on rats with secondary lymphedema

**Sang Ah Kim**[1,2], **Ma. Nessa Gelvosa**[1], **Hwayeong Cheon**[3], **Jae Yong Jeon**[1]*

**1** Department of Rehabilitation Medicine, Asan Medical Center, University of Ulsan College of Medicine, Seoul, Korea, **2** Department of Biomedical Science, Graduate school of University of Ulsan College of Medicine, Seoul, Korea, **3** Biomedical Engineering Research Center, Asan Institute for Life Sciences, Asan Medical Center, Seoul, Korea

* jyjeon71@gmail.com

**Data Availability Statement:** All relevant data are within the paper and its Supporting Information files.

## Abstract

Cancer-related lymphedema (LE) is often caused by radiotherapy and surgery such as lymph node dissection (LND). Previous studies have reported that exercise is beneficial to relieve LE, but the changes in the lymphatic system following exercise are still unclear. This study aimed to examine the changes in lymphatic drainage pathways over the exercise period and beneficial effects of exercise in rats with LE. Twelve rats were randomly allocated into exercise and control groups (EG and CG; n = 6 each). To obtain LE, inguinal and popliteal LND followed by 20 Gy irradiation was performed. Treadmill exercise was 30 minutes/day, 5 days/week over the four-week period. Consecutive indocyanine green (ICG) lymphography images were collected and classified into five patterns: i) linear; ii) splash; iii) stardust; iv) diffuse, and v) none. Ankle thickness was measured weekly. Histopathological evaluation was performed to examine the skin thickness, collagen area fraction (%) and lymphatic vessel density in harvested tissue. ICG lymphography exhibited more linear and splash patterns in the EG at week 3. The difference of swelling between both groups was significantly different at week 4 (p = 0.016). Histopathologic data revealed a thinner epidermis (p = 0.041) and dermis (p = 0.002), lower collagen area fraction (%, p = 0.002), and higher lymph vessel density (p = 0.002) in the EG than the CG. In conclusion, we found that postoperative exercise can facilitate improvement in lymphatic fluid retention in the lymphedema rat model, resulting in improvement of pathological conditions in the lymphatic system.

## Introduction

Cancer-related lymphedema (LE) is a chronic and debilitating disorder of the lymphatic system. Induced by lymphatic dysfunction, LE encompasses problems in lymphatic drainage, pumping and contractility, resulting in the accumulation of protein-abundant fluid in the interstitial space, thereby increasing volume [1, 2]. Secondary LE can be caused by radiation therapy, adjuvant chemotherapy and surgery such as lymph node dissection (LND) for patients with cancer. LE often progresses beyond excessive interstitial fluid accumulation to fibrosis.

**Funding:** JYJ was received fundings from the Asan Institute for Life Sciences, Asan Medical Center, Seoul, Korea (2021IP0037; https://ails.amc.seoul. kr/), and the National Research Foundation of Korea (NRF) grant funded by the Korea government (Ministry of Science and ICT, MSIT) (No. NRF-2019R1A2C1009055; https://www.nrf.re. kr/). The funders had no role in study design, data collection and analysis, decision to publish, or preparation of the manuscript.

**Competing interests:** The authors have declared that no competing interests exist.

There are currently several treatment options for LE. Vascular lymph node transplantation is a surgical option that involves the extraction of the normal lymph nodes (LNs) to transfer to the area affected by LE [3]. Lymphatic venous anastomosis, known as micro-operative, improves lymphatic flow by connecting lymphatic vessels to veins [4]. Non-surgical treatment primarily involves complex or complete decongestive therapy comprising manual lymphatic drainage, skin care, exercise, compression garments, and self-massage, which clinics have employed to relieve LE [5–7].

Exercise has been shown to improve symptoms, quality of life and muscle strength without the aggravation of LE [7, 8]. Previous reports have also suggested that physical exercise after LND can prevent LE occurrence [9]. This may be because extrinsic forces such as skeletal muscle contraction promote lymphatic flow and improve the blood circulation together with lymphatic circulation [10, 11]. ICG lymphography can be used to visualize changes in lymphatic drainage pathways [12, 13]. To better understand the effects of postoperative exercise, clarifying the changes in lymphatic drainage caused by exercise is vital. However, previous studies did not report on drainage changes with postoperative exercise [8, 9]. Our study therefore aims to use the rat hindlimb LE model to show changes in lymphatic drainage patterns with exercise immediately after surgical induction of LE and subsequently determine its contribution to improving pathological conditions in rats.

## Materials and methods

All animal procedures were approved by the Institutional Animal Care and Use Committee (IACUC) of the Asan Institute for Life Sciences, Asan Medical Center, Seoul, Republic of Korea (2020-12-275). The IACUC abides by the relevant guidelines including the ILAR and ARRIVE guidelines and this study strictly followed the recommendations in the IACUC. Fifteen 8-week-old female Sprague–Dawley rats (JA BIO Corporation, Republic of Korea) weighing 250–300 g were used. They were housed in a temperature- and light-controlled room. Twelve rats were randomly assigned to the exercise or control groups (EG and CG; n = 6 per group). Additionally, a subgroup of three rats that had undergone popliteal LN removal only was included to compare with the LE models, which had undergone removal of both popliteal and inguinal LNs.

### Lymphedema induction procedure

To obtain the hindlimb LE model, rats were anesthetized to minimize suffering through inhalation of 4% isoflurane and injection of 10 mL/kg of tiletamine/zolazepam (Zoletil 50, Virbac, France) and xylazine (Rompun, Bayer Korea, Republic of Korea) at a volume ratio of 5:1 mixture before removal of the left inguinal and popliteal LNs. Only popliteal LN was removed in one subgroup. The surgical area was prepared by use of an electronic hair clipper followed by epilation with cream. Evans blue dye solution (30 mg/mL, 30 μL) was injected intradermally into the left side of the base of the tail and left hind paw to stain the inguinal and popliteal LNs, respectively (Fig 1A). Circumferential skin incision to subcutaneous tissue along the left groin was performed, and then the dyed inguinal and popliteal LNs were located. The incision line was above the location of the LNs (Fig 1B), which were extracted together with the surrounding adipose tissue. Edges of the incised skin were cauterized using an electrocautery device (Bovie® Medical Corporation, item No. 18010–00, USA) and sutured to the fascia using 4–0 nylon, leaving a 5 mm gap [14]. A single dose of 20 Gy radiation [15] was delivered into the groin area with an X-ray irradiator (X-Rad 320, Precision, CT, USA) while an 8-mm lead plate covered the rest of the body on postoperative day 2.

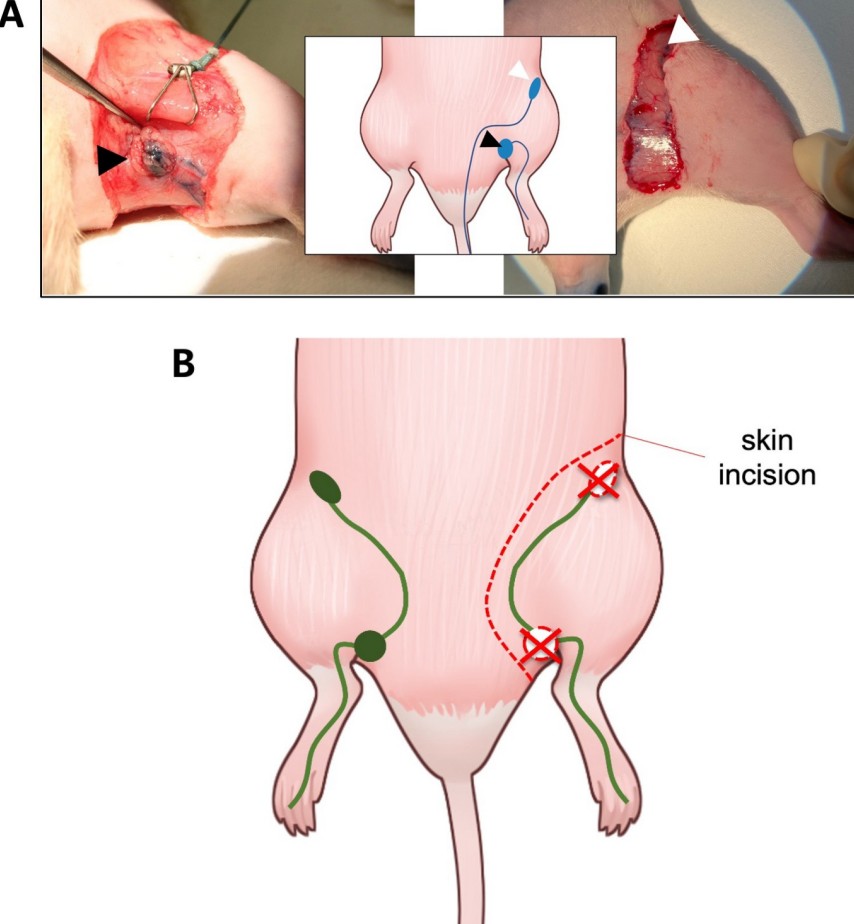

**Fig 1. Schematic illustrating the surgical procedure.** (A) Located LNs, arrowheads indicate popliteal LN (left, black) and inguinal LN (right, white) after injection at hind footpad and base of tail, respectively. (B) Incision line (dashed line) was located above the location of the LNs for operated limb. LN, lymph node.

## Treadmill exercise protocol

The EG started treadmill running 1 week after surgery. Rats were acclimatized to the treadmill (HYB®, LT320, No. 99–0070, China) by gradually increasing the speed over a 5-day period (5 to 12 m/min) throughout the first week of exercise [16]. Thereafter, the EG performed treadmill running at 10–12 m/min, 30 minutes/day, 5 days a week over 4 weeks. The CG rats did not perform any exercise after the operation.

## Hindlimb thickness measurement

Rats were photographed weekly with the hair removed lying on a plate under 4% isoflurane inhalation. Ankle thickness 10 mm above the heel was then measured using ImageJ (Version 1.53, National Institutes of Health, Bethesda, MD, USA). The difference in ankle thickness relative to the contralateral limb was calculated to describe the extent of the swelling, and account for age-appropriate growth and variations in animal size. Analyses were performed by the same investigator blinded to the treatment groups. If the initial volume difference was more than 5%, it was regarded as edema due to lymphedema [15].

## Analysis of ICG lymphography drainage patterns in hindlimb

Lymphatic pattern images were visualized using a customized near-infrared camera with a 730 nm high-power LED and bandpass filter (LST1-01G01-FRD1-00, Opulent Americas, NC, USA; FF01-832/27-50-D, Semrock, NY, USA). Under inhalation anesthesia, rats were injected intradermally with 30 μL ICG solution (1 mg/mL ICG and 2.5 mg/mL bovine serum albumin) into the hind paw. The injected area and hindlimb were massaged to allow the ICG solution to be absorbed into the lymphatic capillaries. Consecutive images just after injuries on ventro-proximal hindlimb were collected, which were then classified according to five patterns (linear, splash with linear, stardust, diffuse, and none) as previously described [17] and scored (1 to 5) as appropriate (Fig 2). This procedure was performed every other week for 5 weeks (three times), starting from the initiation of the exercise program.

## Analysis of histology and immunohistochemistry

Rats were euthanized at week 5 by carbon dioxide asphyxiation. Both hindlimbs were harvested and fixed in 4% buffered formalin for 24 hours at room temperature. Twenty-four samples of hindlimb (10 mm above the heel) were embedded in paraffin and segmented into slices at 5 μm-thick following adequate decalcifications. To measure the epidermal and dermal thickness and collagen area fraction (%, to tissue area below the epidermis) in the harvested tissue, forty-eight specimens were stained with hematoxylin and eosin (H&E) and Masson's trichrome (MT), respectively. Immunohistochemistry (IHC) staining was performed with twelve specimens to visualize lymphatic vessels using mouse monoclonal antibody D2-40 (ACR266B, Biocare Medical, CA, USA), which reacts to podoplanin found in lymphatic endothelium at a dilution of 1: 100.

Using a microscope (BX40, Olympus, Japan) and Olympus cellSens Standard software, a blinded investigator randomly selected four fields in each specimen for analysis and then calculated the following data according to a previously described protocol [18]: 1) thickness of epidermis and dermis in each H&E slide at x200 and x40, respectively; 2) collagen area fraction (%, to tissue area below the epidermis) stained by MT using ImageJ program; and 3) the number of D2-40 positive lymphatic vessels per high-power field (/HPF). Each data accounted for age-appropriate changes in growth and animal size using the following formula: (operated limb–non-operated limb) / (operated limb + non-operated limb).

## Statistical analysis

Data were presented as mean ± standard deviation values and were analyzed using GraphPad Prism (GraphPad Software, Inc., San Diego, CA, USA). The Mann-Whitney U test was used to

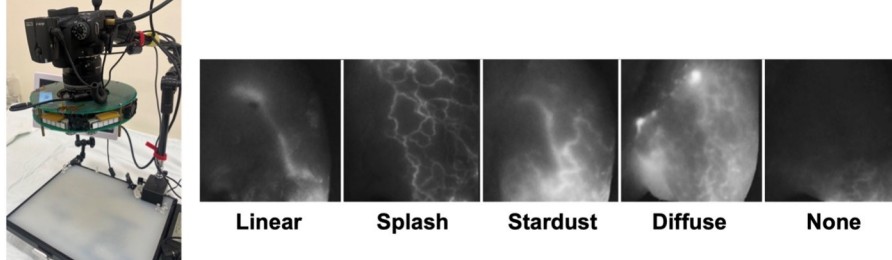

**Fig 2.** Customized camera to visualize lymphatic pattern (left) and classification of lymphatic pattern (right). We classified and scored ICG lymphography images as follows: i) linear, clear lines appear (score 1); ii) splash with linear, the linear lymphatic vessels lie and several linear lymphatic vessels have a winding appearance (score 2); iii) stardust, shining dots are dispersed together with a faint luminous lymphatic vessel (score 3); iv) diffuse, blurry without a clear shape (score 4); and v) none, there is no pattern (score 5).

compare the two groups. The Friedman test was used for comparisons over time, followed by Dunn's multiple comparisons test. The Wilcoxon signed-rank test was used to compare the severity of ICG lymphography patterns over time in both groups. P-values of < 0.05 were considered statistically significant.

## Results

### Hindlimb thickness

As shown in Fig 3A, there was no difference in ankle thickness between the operated and contralateral limbs in the exercise and control groups at week 0. Figures of ankle thickness difference were the greatest at week 1 in EG and CG (1.829 ± 0.655 mm and 2.130 ± 0.464 mm, respectively). A reduction in swelling was observed in rats across both groups; however, a significant difference between the two groups was noted at week 4, with EG having a reduced ankle thickness difference from normal compared with CG (0.312 ± 0.232 mm vs. 0.950 ± 0.460 mm; p = .016, Mann–Whitney U test). Table 1 shows the extent of swelling in the EG decreased significantly at weeks 3, 4 and 5 compared with week 1 (p = .018, .030, and .030, respectively, Dunn's test). Among the rats that underwent popliteal LN removal only, no gross swelling was observed and there was no notable difference in thickness between the ankles over time.

### Changes in ICG lymphography drainage patterns in the hindlimb

Both groups had a similar score distribution at week 1 (EG: 4.2 ± 1.1 vs. CG: 4.0 ± 1.0) (Fig 3B, left). Whereas the EG had a lower severity score than CG at week 3, although the difference was not statistically significant. In the EG, dermal backflow disappeared and showed linear and splash patterns from week 3, while the CG continued to have dermal backflow at week 3 (Fig 3B, right). Moreover, ICG lymphography patterns improved with exercise (p = .094 vs. .375, respectively, Wilcoxon signed-rank test). Rats in the EG had a greater proportion of linear and splash patterns than the CG at week 3 (EG: 3/6 vs. CG: 1/6) (S1 Table). At week 5, the severity score in both groups was almost identical. New pathways were seen beyond the incision line in the groin region toward the axillary region at week 5 (n = 1, Fig A–C in S1 Fig). Rats in the EG that had undergone removal of both popliteal and inguinal LNs exhibited a ventro-cranial collateral pathway in place of the existing lateral pathway, which can be seen in rats that only underwent popliteal lymph node removal (Fig D in S1 Fig).

### Histology and immunohistochemistry

Specimens stained with H&E in the CG showed significant thickening of the epidermis compared with the EG (EG: 0.12 ± 0.06 μm vs. CG: 0.22 ± 0.07 μm, p = .041) and dermis (EG: 0.05 ± 0.03 μm vs. CG: 0.2 ± 0.02 μm, p = .002) (Fig 3C–3E). The collagen area was significantly larger in the CG than in the EG (EG: 0.02 ± 0.01 vs. CG: 0.09 ± 0.03%, p = .002) (Fig 3F and 3G). Additionally, the lymphatic vessel density at the site of LND was significantly higher than the CG (EG: 4.7 ± 0.6 vs. CG: 3.0 ± 0.4 /HPF, p = .002) (Fig 3H and 3I).

## Discussion

We obtained rat hindlimb LE models and demonstrated that postoperative exercise facilitated recovery in improved lymphatic drainage and swelling in the lymphedematous limb. We subsequently observed less fibrous tissues, dermal thickness, and denser tissue lymphatic vessels after the exercise period. These findings indicate that exercise may improve lymphatic fluid retention and temporarily reduce swelling with more persistent effects.

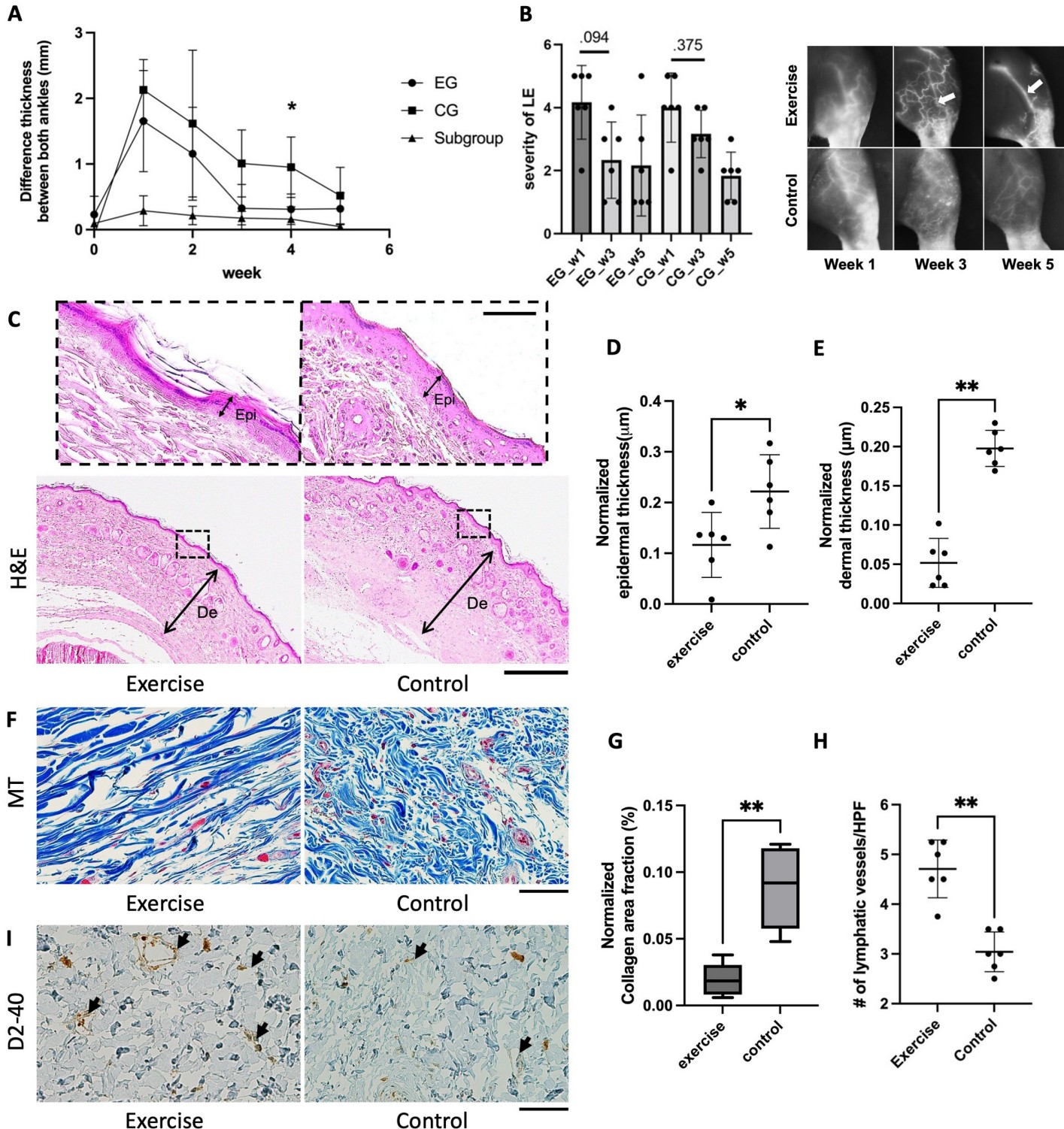

**Fig 3. Comparison of changes in swelling, lymphatic pattern and pathophysiologic analysis between exercise and control groups.** (A) Thickness difference of both ankles in the exercise, control and subgroup measured with ImageJ. The p-values for each time point between exercise and control groups are as follows: week 1, p = .15; week 2, p = .631; week 3, p = .055; week 4, p = .016; and week 5, p = .423 (n = 6 per group; * p < .05, Mann–Whitney U test). Subgroup represents a group of rats that underwent removal of only popliteal LN (n = 3). EG, exercise group; CG, control group; LN, lymph node. (B) Comparison between EG and CG of ICG lymphography patterns on ventroproximal. Data are expressed as mean ± standard deviation. (left) Time course of score on the severity of the patterns in the EG and CG. In the EG, there

were more linear and splash patterns than in the CG. The Wilcoxon signed rank test was performed to compare severity over time in each group. (right) Representative ICG images of the hindlimb at weeks 1, 3, and 5. Arrow indicates a linear signal. EG, exercise group; CG, control group. (C-I) Comparison of skin thickness, fibrosis and the number of LV between both groups. (C) Representative images of H&E-stained skin thickness of the operated leg. (D, E) Control group showed significantly thicker epidermis and dermis compared with the exercise group (n = 6, respectively; p < .05, Mann–Whitney U test). (F) MT staining of operated limbs showed the extent of skin fibrosis. (G) Collagen area fraction (%) in the control group was significantly higher than in the exercise group. (H, I) The number of lymphatic vessels per HPF significantly increased after exercise. Arrows indicate positive podoplanin lymphatic vessels. Scale bar: C = 50 μm and 200 μm, respectively; F and I = 50 μm. Original magnification: C = × 200 and × 40, respectively; F and I = × 400. Epi, epidermis; De, dermis; LV, lymphatic vessel; H&E, hematoxylin and eosin; MT, Masson's trichrome; HPF, high-power field. * < .05, ** < .01.

Previous researchers studied the creation of a hindlimb LE rat model [14, 15, 19]. Recently, Huang CW et al. [20] established a successful hindlimb LE rat model by removing popliteal and inguinal LN with a circumferential incision to block collateral lymphatic vessels. In our study, the subgroup that underwent the removal of popliteal LN showed mild swelling compared with the CG. Most rats with the removal of popliteal LN have a collateral pathway toward axillary LN through alongside the internodal blood vessel [21, 22]. However, we found the ventro-cranial collateral pathway after removing both the inguinal and popliteal LN, similar to the observations by Takeno Y et al. [23], which may explain the long-lasting model due to a lengthier time to detour using this pathway. Our findings suggest removing both popliteal and inguinal LNs, and to include a circumferential incision to ensure relatively prolonged hindlimb LE models.

Using ICG lymphography, a non-invasive method to identify the integrity of lymphatic function [12, 13, 23, 24], we traced changes in the lymphatic drainage system, including collateral pathways and dermal backflow. Dermal backflow represents aggravation of abnormal lymphatic drainage. Takeno et al. found that rats without surgery exhibited a linear signal in the hindlimb, while those that underwent inguinal and popliteal lymph node dissection showed dermal backflow patterns, such as stardust [12]. Our data showed that the dermal backflow pattern in the EG rats tended to decrease 2 weeks after beginning treadmill exercise compared with the CG rats. Additionally, a significant reduction in swelling of the lymphedematous hindlimb in the EG was observed from week 3, which differed significantly from that of the CG at week 4. Considering the recovery speed, we can assume that rats in the EG solved the drainage obstruction earlier than the CG.

After exercise, we found newly activated lymph vessels toward axillary LN from inguinal LN (n = 1) beyond the incision line. Previous reports suggest that the major compensatory pathway from the hindlimb after disruption of the existing pathway is toward inguinal and

**Table 1. Thickness difference between both ankles for each week.**

|  | Exercise | Control | *P-value* |
|---|---|---|---|
| Week 1 | 1.829 ± 0.655 | 2.130 ± 0.464 | 0.15 |
| Week 2 | 1.155 ± 0.708 | 1.617 ± 1.118 | 0.631 |
| Week 3 | 0.325 ± 0.363[†] | 1.008 ± 0.510 | 0.055 |
| Week 4 | 0.312 ± 0.232[†] | 0.950 ± 0.460 | 0.016* |
| Week 5 | 0.319 ± 0.242[†] | 0.519 ± 0.430[†] | 0.423 |
| Friedman | < .001[a] | 0.002[a] |  |
| *P-value* |  |  |  |

Data are shown as mean ± standard deviation.

[a] Significantly different among weeks (P< 0.05, Friedman test)

[†] Significantly different between week 1 and following weeks (P< 0.05, Dunn's post-hoc test)

* Significantly different between groups for each week (P< 0.05, Mann–Whitney U test)

axillary LNs [12, 21, 25]. Additionally, we observed higher lymphatic vessel density after the exercise protocol. Podoplanin with antibody D2-40 is an important marker of endothelium that can distinguish collecting lymphatic vessels [26, 27] and is relatively specific to lymphatic vessels compared with other lymphatic endothelial markers [28]. These findings can suggest that exercise facilitates recruitment of previously inactive lymph vessels and lymphangiogenesis.

The rapid decrease in swelling after exercise may be attributed to lymphatic circulation and preserved lymphatic function together with muscle pumping associated with exercise. Additionally, lymphatic flow may be associated with exercise-induced muscle contraction [7, 29]. The flow is managed by extrinsic and intrinsic forces such as the skeletal and smooth muscles contraction in the vessels, respectively, which is accelerated by physical activity [10, 11]. The intact valvular function requires unidirectional continuous lymphatic flow wherein skeletal muscle plays a crucial role as an extrinsic force. In addition, several studies have reported that physical activity can induce a myokine response including irisin, which is produced by skeletal muscle activation [30–32]. This response leads to anti-inflammation, which may alleviate lymphatic inflammation. Therefore, it is plausible that physical activity may lead to an accumulation of myokine, which may explain the mechanisms that promote the positive effects of exercise on lymphatics. Taken together with the results of our ICG lymphography, our hypothesis that exercise may provide the propulsion of lymphatic flow, facilitating the recovery of lymphatic fluid retention, is supported. Further studies regarding the mechanisms of irisin are needed.

The H&E slide analysis show a thicker epidermis and dermis in the CG compared with the EG. We also observed a larger collagen area fraction (%) with skin fibrosis in the CG. The relationship between dermal thickness and LE severity has been previously reported [27, 33, 34]. Skin thickening may result from excessive accumulation of lymphatic fluid in the cutaneous layer (i.e., dermal backflow) with the refluxed lymphatic fluid becoming adipose tissue. Similarly, a recent study with H&E staining observed thicker epidermis in rats with LE compared with the sham group [20]. Our findings conclude that postoperative exercise may lower the occurrence of fibrosis, which can be caused by relatively severe swelling and long-lasting lymphatic fluid retention.

## Limitations

There are some limitations to the current study. First, achieving statistical significance and further translating findings to humans is challenging given the limited sample sizes. Second, despite similar lymphatic structures, the rat LE model is not a perfect representation of the human body given their ability to recover faster from lymphatic system obstruction. Nevertheless, an experimental study should be conducted for at least 4 weeks to identify the role of exercise, as demonstrated in previous animal studies using treadmill exercise. This study used long-lasting LE models for up to 5 weeks and showed the therapeutic effect of exercise. Future studies could investigate prophylactic effect of exercise in more rodent models with exercise before surgery and radiation therapy.

## Conclusions

This study found that postoperative exercise can expedite recovery of lymphatic drainage and relieve swelling, thereby reducing the occurrence of fibrosis in rats that underwent removal of popliteal and inguinal lymph nodes followed by radiotherapy. However, further studies are required to elucidate the mechanism.

## Supporting information

**S1 Table. Classification of ICG pattern in both groups.**
(DOCX)

**S1 Fig. Visualized lymphatic pathways with ICG lymphography.**
(DOCX)

**S1 Appendix. The result of Dunn's post-hoc test for changes in swelling over time and normality test for each data.**
(XLSX)

## Acknowledgments

We thank the core facilities of the Comparative Pathology Laboratory and Animal Experiment Laboratory at the ConveRgence mEDIcine research center (CREDIT), Asan Medical Center, for sharing their equipment, services, and expertise with us. Additionally, we would like to express our gratitude to Dr. Linhai Chen for his excellent assistance during the rat surgery.

## Author Contributions

**Conceptualization:** Sang Ah Kim, Ma. Nessa Gelvosa, Hwayeong Cheon, Jae Yong Jeon.

**Data curation:** Sang Ah Kim.

**Formal analysis:** Sang Ah Kim.

**Funding acquisition:** Hwayeong Cheon, Jae Yong Jeon.

**Investigation:** Sang Ah Kim.

**Methodology:** Sang Ah Kim, Ma. Nessa Gelvosa, Hwayeong Cheon, Jae Yong Jeon.

**Project administration:** Jae Yong Jeon.

**Supervision:** Jae Yong Jeon.

**Validation:** Sang Ah Kim.

**Visualization:** Sang Ah Kim.

**Writing – original draft:** Sang Ah Kim.

**Writing – review & editing:** Sang Ah Kim, Ma. Nessa Gelvosa, Hwayeong Cheon, Jae Yong Jeon.

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
