## [Decision Letter · Decision Letter 0]

29 Dec 2022

PONE-D-22-30319The effects of postoperative treadmill exercise on rats with secondary lymphedemaPLOS ONE

Dear Dr. Jeon,

Thank you for submitting your manuscript to PLOS ONE. After careful consideration, we feel that it has merit but does not fully meet PLOS ONE’s publication criteria as it currently stands. Therefore, we invite you to submit a revised version of the manuscript that addresses the points raised during the review process.

We look forward to receiving your revised manuscript.

Kind regards,

Mehmet Cudi Tuncer, Ph.D.

Academic Editor

PLOS ONE

Journal Requirements:

Reviewers' comments:

Reviewer's Responses to Questions

**Comments to the Author**

1. Is the manuscript technically sound, and do the data support the conclusions?

Reviewer #1: Yes

Reviewer #2: No

2. Has the statistical analysis been performed appropriately and rigorously? 

Reviewer #1: Yes

Reviewer #2: Yes

3. Have the authors made all data underlying the findings in their manuscript fully available?

Reviewer #1: Yes

Reviewer #2: Yes

4. Is the manuscript presented in an intelligible fashion and written in standard English?

Reviewer #1: Yes

Reviewer #2: Yes

5. Review Comments to the Author

Reviewer #1: This manuscript aims to provide badly needed information about exercise effects on lymphedema. The authors use a rat model, which they recognize as limited in ability to accurately portray human lymphedema. Time points were chosen to capture effects that appear to wane by 5 weeks post-RT. At 3 weeks post-RT, however, significant differences were noted in EG and CG.

Several studies by other groups have shown that exercise, and some studies have shown that irisin, which is produced in exercise, have positive effects on lymphatics--these studies should be mentioned in the discussion and cited. doi: 10.1113/JP271757, doi: 10.1155/2021/6572342, doi: 10.1096/fj.201800178R.

Why not exercise some rats before surgery/RT to investigate prophylactic effect of exercise? (remove lines 281-284, replace with "Future studies could investigate prophylactic.....")

Figure 3B shows significance, but only at week 3--this is probably due to the difficulty of deriving an accurate animal model of RT.

The analysis using ICG is subjective and difficult to judge from Figure 3B. EG at week 5 appears to have some backflow, and CG did not take up dye--this figure should be enlarged and include more than images from just one rat per time point, or deleted entirely--the fibrosis, collagen fraction, and # of lymphatic vessels/HPF are more convincing. Figure 3C H&E skin thickness appears to have selected thickest area for CG and thinner area for EG--maybe use a different example set for these images?

Reviewer #2: The author describes one of the first experimental animal model for Lymphedema. At the state of art, there is no recognized in vivo model for the study of lymphedema, although recent reports have proposed a rat model. The authors developed the model previously described and conducted a study assessing the effect of physical exercise on surgically induced lymphedema.

This is an interesting and original study. Nevertheless, there are some concerns that should be addressed before considering the manuscript for publication.

The authors propose a new animal model for lymphedema. Have you validated/ described it before conducting this experimental study? Which criteria did you use to make sure that all the animals developed lymphedema, other than postoperative swelling? Do you have a baseline of the ICG study to support its use in your model?

The authors operated one limb of each animal and used the contralateral as a control for at least the measurement of the circumference/thickness of the limbs. They correctly used the difference of the thickness between the healthy limb and the operated one to compare experimental and control group. It is not clear in the manuscript if they use the contralateral healthy limb as control also for the comparison of lymphatic pattern, histology and immunohistochemistry between exercise and control groups. The exercise itself along with control of weight could induce reduction of the thickness of the tissue and influence the composition of connective tissue and tissue markers even in healthy limb. I think that to properly evaluate the effect of the exercise on lymphedema, the data of the affected limbs should be related to those from the healthy limb.

6. PLOS authors have the option to publish the peer review history of their article (what does this mean?). If published, this will include your full peer review and any attached files.

Reviewer #1: **Yes: **Melissa B Aldrich

Reviewer #2: No

---

## [Author Response · Author response to Decision Letter 0]

2 Feb 2023

Responses to Reviewer 1

Comment 1: This manuscript aims to provide badly needed information about exercise effects on lymphedema. The authors use a rat model, which they recognize as limited in ability to accurately portray human lymphedema. Time points were chosen to capture effects that appear to wane by 5 weeks post-RT. At 3 weeks post-RT, however, significant differences were noted in EG and CG.

Response: Thank you for your understanding of our manuscript. As you have mentioned, this manuscript aimed to provide beneficial information about the effects of exercise on lymphedema. In this study, radiation therapy was used as a method to induce chronic lymphedema together with lymph node dissection to reproduce human secondary lymphedema. Despite the functional defect of the lymphatic vessels due to external injury, the exercise group tended to have less lymphedema than the control group from immediately after surgery to the 5th week. The difference between the groups was most prominent at week 3, which can be seen as promoting the functional recovery of lymph through exercise performance. This can be supported by our results of histology/IHC as well as ICG lymphography at 3 weeks.

 In animal models, considering the turnover rate of interstitial fluid and lifespan compared to humans, it cannot be expressed in the same cycle as humans (Dutta, Sulagna, and Pallav Sengupta. “Men and mice: Relating their ages.” Life sciences vol. 152 (2016): 244-8. doi:10.1016/j.lfs.2015.10.025). We believe that reproducing human chronic lymphedema is limited as animal models tend to have a faster rate of disease development and recover faster than that of humans. However, animal studies may be valuable for studying the progression and treatment of long-lasting chronic diseases with relatively short observations. Therefore, the current study is meaningful in providing evidence for future studies to prove that exercise can be effective for patients with lymphedema. We have described this in the Discussion section on page 12, lines 255 to 256 and the Limitation section on page 14, lines 290 to 294.

Comment 2: Several studies by other groups have shown that exercise, and some studies have shown that irisin, which is produced in exercise, have positive effects on lymphatics--these studies should be mentioned in the discussion and cited. doi: 10.1113/JP271757, doi: 10.1155/2021/6572342, doi: 10.1096/fj.201800178R.

Response: We are grateful for the insightful comments and suggestions helpful to our manuscript. This made us look at our findings in a different light. However, in the case of our study, it seems slightly out of scope because exercise-induced irisin response has been associated with obesity and mesenteric lymphatic function in previous studies. The mechanism remains unclear in terms of lymphatic system in the extremities. Nonetheless, your suggestion has been included in the Discussion section as a possible mechanism to direct future research. It can be found on page 13, lines 271 to 275.

Comment 3: Why not exercise some rats before surgery/RT to investigate prophylactic effect of exercise? (remove lines 281-284, replace with "Future studies could investigate prophylactic.....")

Response: We have modified our manuscript taking into account your comments. This change can be found on page 14, Limitation, lines 294 to 296.

Comment 4: Figure 3B shows significance, but only at week 3--this is probably due to the difficulty of deriving an accurate animal model of RT.

Response: As I mentioned in my response to comment 1, we induced lymphedema by performing lymph node dissection along with RT. We therefore considered the animal models to be accurate for lymphedema according to our criterion that a volume difference greater than 5% was indicative of lymphedema as previous study mentioned. We have updated this information on page 6, lines 118 to 119. In addition, we observed that the rats in the exercise group had more severe swelling compared to those of control group at week 4. Moreover, as the lymphangiography (Figure 3B) showed significance at week 3, we were able to say that the improved lymphatic flow was associated with significantly lower ankle thickness at week 4.

Comment 5: The analysis using ICG is subjective and difficult to judge from Figure 3B. EG at week 5 appears to have some backflow, and CG did not take up dye--this figure should be enlarged and include more than images from just one rat per time point, or deleted entirely--the fibrosis, collagen fraction, and # of lymphatic vessels/HPF are more convincing. 

Response: Thank you for pointing this out. Although the ICG lymphography may be subjective, it is a method primarily used to investigate accessory and/or collateral lymphatic pathways as well as to evaluate lymphatic function in vivo without sacrificing rats. As described in the manuscript, the same amount of dye was injected into each rat and relatively reduced lymphatic function is expected to result in poor drainage of the dye in the CG. However, we agree with your comment and have replaced the figures in Figure 3B with others to clarify our intentions. 

Comment 6: Figure 3C H&E skin thickness appears to have selected thickest area for CG and thinner area for EG--maybe use a different example set for these images?

Response: We have provided alternative figures in Figure 3C based on your suggestion.

Responses to Reviewer 2

Comment 1: The author describes one of the first experimental animal model for Lymphedema. At the state of art, there is no recognized in vivo model for the study of lymphedema, although recent reports have proposed a rat model. The authors developed the model previously described and conducted a study assessing the effect of physical exercise on surgically induced lymphedema.

This is an interesting and original study. Nevertheless, there are some concerns that should be addressed before considering the manuscript for publication.

Response: Thank you for your comprehensive understanding of our manuscript. However, the rat model used in this study is not the first experimental animal model for lymphedema, and we tried to derive an optimal model based on the lymphedema rodent model as previous studies had described and use it in the current study. We have mentioned this in the Discussion section on page 12, lines 239 to 247.

Please note that the following articles on the experimental animal model for lymphedema:

1. Kanter MA, Slavin SA, Kaplan W. An experimental model for chronic lymphedema. Plast Reconstr Surg. 1990;85(4):573–580. doi:10.1097/00006534-199004000-00012

2. Yang CY, Nguyen DH, Wu CW, et al. Developing a lower limb lymphedema animal model with combined lymphadenectomy and low-dose radiation. Plast Reconstr Surg Glob Open. 2014;2(3):e121. Published 2014 Apr 7. doi:10.1097/GOX.0000000000000064

3. Huang CW, Chang YC, Hsiao HY, Liu JW, Chang FC, Huang JJ. Creation of a rat lymphedema model using extensive lymph node dissection and circumferential soft tissue resection: Is this a reliable model?. Microsurgery. 2021;41(8):762–771. doi:10.1002/micr.30817

Comment 2: The authors propose a new animal model for lymphedema. Have you validated/ described it before conducting this experimental study?

Response: We have described the content about animal model in the Discussion section on page 12, lines 244 to 248. The animal model has been reported in previous studies and we identified that the model with removal of both inguinal and popliteal lymph nodes was validated compared to removal of popliteal lymph node alone.

Comment 3: Which criteria did you use to make sure that all the animals developed lymphedema, other than postoperative swelling? 

Response: Postoperative edema (surgery-related edema) rather than lymph node dissection may be observed at week 1. However, in a previous study, edema volume caused by sham surgery was significantly less than that of lymphedema, and a linear pattern with lymph nodes was captured by ICG lymphography (Huang CW et al. 2021, among the list of studies above). This is supported by our results of week 2 volume, which increased to more than the week 1 volume, and the ICG lymphography. An explanation of the criteria has been added on page 6, lines 118 to 119.

Comment 4: Do you have a baseline of the ICG study to support its use in your model?

Response: ICG lymphography can be used to investigate lymphatic function in vivo. ICG staging has also been applied for assessing severity of lymphedema in humans. Additionally, our previous study showed significant results in ICG lymphography staging, which can represent the severity of dermal backflow according to changes in lymphatic function (Cheon H. et al. 2022). In mild stages of lymphatic dysfunction, lymphatic vessels appear as several distinct lines, either extended to find new pathways or alternative pre-existing pathways. This is called the splash pattern. Lymphatic drainage is not properly performed, and as lymphatic function gradually deteriorates, back flow is formed on the surface, which shows several points shining like stars, known as stardust. In a more serious situation, the lymphatic fluid does not take the form of a lymphatic vessel, and the lymphatic fluid spreads to several places and appears blurry, which is called a diffuse pattern. This content can be found on page 6, lines 127 to 129 and the legend in Figure 2.

Please refer to the following previous studies: 

1. Cheon, H., Gelvosa, M. N., Kim, S. A., Song, H. Y., & Jeon, J. Y. (2022). Lymphatic channel sheet of polydimethylsiloxane for preventing secondary lymphedema in the rat upper limb model. Bioengineering & Translational Medicine, e10371. http://dx.doi.org/10.1002/btm2.10371

2. Yamamoto T, Narushima M, Doi K, et al. Characteristic indocyanine green lymphography findings in lower extremity lymphedema: the generation of a novel lymphedema severity staging system using dermal backflow patterns. Plast Reconstr Surg. 2011;127(5):1979–1986. doi:10.1097/PRS.0b013e31820cf5df

Comment 5: The authors operated one limb of each animal and used the contralateral as a control for at least the measurement of the circumference/thickness of the limbs. They correctly used the difference of the thickness between the healthy limb and the operated one to compare experimental and control group. It is not clear in the manuscript if they use the contralateral healthy limb as control also for the comparison of lymphatic pattern, histology and immunohistochemistry between exercise and control groups. The exercise itself along with control of weight could induce reduction of the thickness of the tissue and influence the composition of connective tissue and tissue markers even in healthy limb. I think that to properly evaluate the effect of the exercise on lymphedema, the data of the affected limbs should be related to those from the healthy limb.

Response: Thank you for pointing this out. We agree with your comment and have revised the method for analysis of histology. To be specific, the comparison of operated limbs between both groups has been updated with the comparison of normalized data between the groups as follows: (operated – non-operated) / (operated + non-operated). This has been mentioned on page 8, lines 155 to 157. Each normalized raw data regarding skin thickness and collagen area can be found in Figure 3D-G and Supporting Appendix S1.

---

## [Decision Letter · Decision Letter 1]

4 Apr 2023

PONE-D-22-30319R1The effects of postoperative treadmill exercise on rats with secondary lymphedemaPLOS ONE

Dear Dr. Jeon,

Thank you for submitting your manuscript to PLOS ONE. After careful consideration, we feel that it has merit but does not fully meet PLOS ONE’s publication criteria as it currently stands. Therefore, we invite you to submit a revised version of the manuscript that addresses the points raised during the review process.

We look forward to receiving your revised manuscript.

Kind regards,

Shimpei Miyamoto

Academic Editor

PLOS ONE

Journal Requirements:

Reviewers' comments:

Reviewer's Responses to Questions

**Comments to the Author**

1. If the authors have adequately addressed your comments raised in a previous round of review and you feel that this manuscript is now acceptable for publication, you may indicate that here to bypass the “Comments to the Author” section, enter your conflict of interest statement in the “Confidential to Editor” section, and submit your "Accept" recommendation.

Reviewer #1: All comments have been addressed

Reviewer #2: (No Response)

2. Is the manuscript technically sound, and do the data support the conclusions?

Reviewer #1: Yes

Reviewer #2: Yes

3. Has the statistical analysis been performed appropriately and rigorously? 

Reviewer #1: Yes

Reviewer #2: Yes

4. Have the authors made all data underlying the findings in their manuscript fully available?

Reviewer #1: Yes

Reviewer #2: Yes

5. Is the manuscript presented in an intelligible fashion and written in standard English?

Reviewer #1: Yes

Reviewer #2: Yes

6. Review Comments to the Author

Reviewer #1: all concerns addressed--no further revisions needed

all figures updated, all limitations and suggestions incorporated

Reviewer #2: I wish to thank the Authors for providing an updated version of the manuscript. They carefully revised the manuscript and he quality of the study improved significantly.

I still would like to receive a clear answer regarding the following point:

“Do you have a baseline of the ICG study to support its use in your model?” The Reviewers and the readers know the staging of lymphedema according to ICG findings in humans. And this is not applicable to other models unless validated. As this imaging technique has not been extensively performed on animal models, authors are encouraged to have a baseline to compare with postoperative findings.

7. PLOS authors have the option to publish the peer review history of their article (what does this mean?). If published, this will include your full peer review and any attached files.

Reviewer #1: **Yes: **Melissa B Aldrich

Reviewer #2: No

---

## [Author Response · Author response to Decision Letter 1]

7 Apr 2023

Responses to editorial comments 

Thank you for providing us with the opportunity to revise our manuscript. We have carefully considered the comments and suggestions of the reviewers and have made revisions accordingly. We hope that these changes and responses meet the expectations of PLOS ONE, and that the manuscript is now suitable for publication.

Comment: If applicable, we recommend that you deposit your laboratory protocols in protocols.io to enhance the reproducibility of your results. Protocols.io assigns your protocol its own identifier (DOI) so that it can be cited independently in the future. For instructions see: https://journals.plos.org/plosone/s/submission-guidelines#loc-laboratory-protocols. Additionally, PLOS ONE offers an option for publishing peer-reviewed Lab Protocol articles, which describe protocols hosted on protocols.io. Read more information on sharing protocols at https://plos.org/protocols?utm_medium=editorial-email&utm_source=authorletters&utm_campaign=protocols.

Response: Thank you for the suggestion. While we appreciate it, we anticipate that our current laboratory protocols will be refined in the near future to yield clearer results. As part of our efforts to enhance the reproducibility of our research findings, we plan to deposit our improved laboratory protocols on Protocols.io.

Response to Reviewer 2

Thank you for reviewing our manuscript. Our answers to your queries are as follows.

Comment 1: I still would like to receive a clear answer regarding the following point:

“Do you have a baseline of the ICG study to support its use in your model?” The Reviewers and the readers know the staging of lymphedema according to ICG findings in humans. And this is not applicable to other models unless validated. As this imaging technique has not been extensively performed on animal models, authors are encouraged to have a baseline to compare with postoperative findings.

Response: Thank you for bringing this to our attention. Previous studies have utilized rats in order to observe lymphatic drainage patterns via ICG lymphography. Ogata et al. analyzed ICG images of hindlimbs from rats that had undergone lymph node dissection seven days after surgery. Their results indicated that the ICG images displayed diffuse or blurry patterns. Additionally, Takeno et al. reported that in a hindlimb non-operative rat model, a linear signal was observed, while in the operative hindlimb after inguinal and popliteal lymph node dissection, dermal backflow patterns, such as stardust, were shown. This suggests that detecting the linear pattern earlier in the operated limb will result in a quicker recovery process and a return to the state prior to surgery. We have described these findings on page 12, lines 249 to 254 in the main text.

Please refer to the list of previous studies I mentioned below:

1. Ogata F, Azuma R, Kikuchi M, Koshima I, Morimoto Y. Novel lymphography using indocyanine green dye for near-infrared fluorescence labeling. Ann Plast Surg. 2007 Jun;58(6):652-5. doi: 10.1097/01.sap.0000250896.42800.a2. PMID: 17522489.

2. Takeno Y, Fujimoto E. Alterations of lymph flow after lymphadenectomy in rats revealed by real time fluorescence imaging system. Lymphology. 2013 Mar;46(1):12-9. PMID: 23930437.

Additional clarifications

In addition to the above comments, all spelling and grammatical errors have been corrected. Furthermore, references 3 (page 15, lines 316 – 318) and 7 (page 16, lines 331 – 332) have been updated with the latest information.

Thank you again for reviewing our manuscript in detail and providing helpful comments. We hope that our responses and the corresponding revisions are satisfactory.

---

## [Editor Report · Decision Letter 2]

24 Apr 2023

The effects of postoperative treadmill exercise on rats with secondary lymphedema

PONE-D-22-30319R2

Dear Dr. Jeon,

We’re pleased to inform you that your manuscript has been judged scientifically suitable for publication and will be formally accepted for publication once it meets all outstanding technical requirements.

Kind regards,

Shimpei Miyamoto

Academic Editor

PLOS ONE
---

## [Editor Report · Acceptance letter]

16 May 2023

PONE-D-22-30319R2 

The effects of postoperative treadmill exercise on rats with secondary lymphedema 

Dear Dr. Jeon:

I'm pleased to inform you that your manuscript has been deemed suitable for publication in PLOS ONE. Congratulations! Your manuscript is now with our production department. 

Kind regards, 

on behalf of

Dr. Shimpei Miyamoto 

Academic Editor

PLOS ONE